# Research Progress on Sulfur Deactivation and Regeneration over Cu-CHA Zeolite Catalyst

**Jiangli Ma** [1,2], **Shiying Chang** [2,3], **Fei Yu** [2,3], **Huilong Lai** [1,2] **and Yunkun Zhao** [1,2,*]

1. State Key Laboratory of New Technologies for Comprehensive Utilization of Rare and Precious Metals, Kunming Institute of Precious Metals, Kunming 650106, China
2. Kunming Sino-Platinum Metals Catalyst Co., Ltd., Kunming 650106, China
3. Sino-Platinum Metals Catalyst (Dongying) Co., Ltd., Dongying 257000, China
* Correspondence: yk.zhao@spmcatalyst.com; Tel.: +86-0871-6839-3370

**Abstract:** Benefiting from the exceptional selective catalytic reduction of NOx with ammonia ($NH_3$-SCR) activity, excellent $N_2$ selectivity, and superior hydrothermal durability, the $Cu^{2+}$-exchanged zeolite catalyst with a chabazite structure (Cu-CHA) has been considered the predominant SCR catalyst in nitrogen oxide (NOx) abatement. However, sulfur poisoning remains one of the most significant deterrents to the catalyst in real applications. This review summarizes the $NH_3$-SCR reaction mechanism on Cu-CHA, including the active sites and the nature of hydrothermal aging resistance. On the basis of the $NH_3$-SCR reaction mechanism, the review gives a comprehensive summary of sulfate species, sulfate loading, emitted gaseous composition, and the impact of exposure temperature/time on Cu-CHA. The nature of the regeneration of sulfated catalysts is also covered in this review. The review gives a valuable summary of new insights into the matching between the design of $NH_3$-SCR activity and sulfur resistance, highlighting the opportunities and challenges presented by Cu-CHA. Guidance for future sulfur poisoning diagnosis, effective regeneration strategies, and a design for an efficient catalyst for the aftertreatment system (ATS) are proposed to minimize the deterioration of NOx abatement in the future. Finally, we call for more attention to be paid to the effects of $PO_4^{3-}$ and metal co-cations with sulfur in the ATS.

**Keywords:** selective catalytic reduction; Cu-CHA; active site; aftertreatment system; sulfur poisoning; sulfur regeneration

## 1. Introduction

With the aggravation of the global energy crisis, lean-burn combustion diesel engines featuring strong power and economical merits have been widely used in road and off-road machines. Excessive air intake during engine combustion can not only increase fuel combustion efficiency but also reduce the contents of carbon monoxide (CO), hydrocarbons (HC), and particulate matter (PM) in exhaust emissions [1]. However, lean burn combustion tends to emit excessive NOx and ammonia ($NH_3$), which cause damage to the environment and human health. In recent years, stringent regulations have been formulated to control exhaust emissions in various countries and regions. In order to comply with these stringent regulations, the ATS, as shown in Figure 1, consisting of a diesel oxidation catalyst (DOC), a diesel particulate filter (DPF), ammonia selective catalytic reduction ($NH_3$-SCR), and an ammonia slip catalyst (ASC), has been successfully developed as a promising technology for NOx emission control [1–4]. The implemented CHINA VI emission regulation requires an 80% reduction in NOx and a 40% increase in durability compared with CHINA V in heavy-duty vehicles [4,5]. Therefore, advanced ATS catalysts shall have better hydrothermal aging resistance and NOx abatement ability, especially during the cold start (below 200 °C) and DPF active regeneration operations (above 650 °C) [5,6]. As can be seen, during DPF active regeneration, excessive fuel is injected into the upstream DOC for soot combustion. This

requires the SCR catalysts to have superior stability to withstand HC adsorption and HTA resistance in humid and high-temperature environments [6].

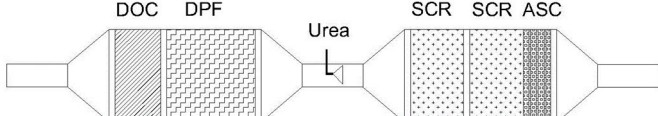

**Figure 1.** A typical advanced emission control system in diesel vehicles.

Recently, Cu-CHA has been reported as one of the most promising commercial catalysts in the current emission control of diesel engines [6–8]. CHA zeolites are a group of small-pore structures (pore size of ~3.8 Å) that consist of connected $TO_4$ (T represents the framework atom) tetrahedral units by sharing oxygen atoms [6]. Because of the advantages of excellent shape selectivity, hydrothermal aging (HTA) resistance, adjustable acid sites, and a huge specific surface, Cu-SSZ-13 has been widely used in the ATS [6,7].

Rapidly expanding regulations have been enabled by the ultra-low sulfur content in fuel [7]. A drastic reduction in sulfur content has been achieved in diesel engines [7,8]. However, even with the ultra-low sulfur content (10 ppm) under CHINA VI regulations, sulfur poisoning remains one of the most significant deterrents to the catalyst in real applications [9,10]. Compared with Fe-zeolite, Cu-zeolite is more sensitive to sulfur, which has been proven by many groups [9–11]. Meanwhile, accumulated lifetime exposure to sulfur in the diesel ATS always results in a dramatic deterioration of NOx conversion [12–14].

Numerous studies have been carried out to explore the damaging effects of sulfur on Cu-CHA structures and the nature of activity recovery during DPF active regeneration [12]. It is generally considered that the deactivation over Cu-CHA is caused by damage to zeolite structures, including the drastic decrease in the specific surface area and the binding of $SO_x$ to copper species [15]. In addition, $SO_2$ and $NH_3$ tend to adsorb together to form sulfate species on isolated $Cu^{2+}$, which further aggravates the pore block and SCR reaction at low temperatures [14,15]. Gao et al. [15] proved that with an increase in $SO_2$ concentration and sulfur exposure time, more sulfate formed and NOx conversion worsened for Cu-SAPO-34. Meanwhile, the active site of $Cu(OH)^+$ can also form copper sulfate species under the impact of $SO_2$. CuO species could further promote the oxidation of $SO_2$ and thus accelerate sulfur poisoning [15,16].

Currently, there are no on-board diagnostics (OBD) specified for sulfur poisoning and regeneration in real applications, resulting in massive parts of failure caused by sulfur poisoning/regeneration. It is essential to explore the nature of sulfur poisoning and regeneration over Cu-CHA, thereby providing guidance for future sulfur poisoning diagnosis, effective regeneration strategies, and a design for an efficient catalyst for the ATS to minimize NOx deterioration in applications.

## 2. Research on Copper's Active Sites and NH₃-SCR Reaction over Cu-CHA

A major breakthrough was achieved several years ago: researchers discovered that Cu-CHA (e.g., Cu-SAPO-34 and Cu-SSZ-13) could meet the HTA resistance and low/high-temperature activity for commercial utilization [15]. Both catalysts were used in the ATS of diesel vehicles before 2011. However, Cu-SAPO-34 experienced unexpected failure in diesel after-treatment commercial utilization due to a lack of durability at low temperatures upon contact with moisture [16]. Wang et al. [16] studied the deactivation mechanism of Cu-SAPO-34 at an atomic level. It was proposed that Cu-SAPO-34 was deactivated by the formation of Cu-aluminate-like species that were derived from the interaction between $Cu(OH)_2$ and the hydrolysis of framework Al species. Compared with Cu-SAPO-34, Cu-SSZ-13 was reported to have better HTA resistance and predominant NO conversion efficiency [17]. Studies further verified the hydrothermal durability of Cu-SSZ-13 and found nearly no deterioration in comparison with the activity of fresh samples [18].

In principle, there are four most probable ion exchange sites on Cu-SSZ-13, i.e., in the center of double-6-membered rings, inside the CHA cage near the face of the six-membered

ring, and in the CHA cage along the three-dimensional axis and near the eight-membered ring [19]. On the basis of previous studies, Gao et al. [20] calculated and analyzed the possible locations of $Cu^{2+}$ active sites on Cu-SSZ-13 by electron paramagnetic resonance (EPR) and other measurements, as shown in Figure 2. The results revealed that $Cu^{2+}$ was able to migrate within different locations and even within unit cells. When the ion exchange rate was lower than 23%, which was normally accompanied by a low Cu loading and a Cu-Cu spacing greater than 20 Å, $Cu^{2+}$ occupied a six-square cell and resided in a six-membered ring. With an increase in the ion exchange rate, Cu loading became relatively higher. $Cu^{2+}$ resided in the six-membered ring, and the other resided near the eight-membered ring in the cage. When the Cu-Cu spacing narrowed to 4 Å–5 Å, the $Cu^{2+}$ was supposed to be located in sites A and B (Figure 2) or the hexagonal cell with a distribution of mirror symmetry.

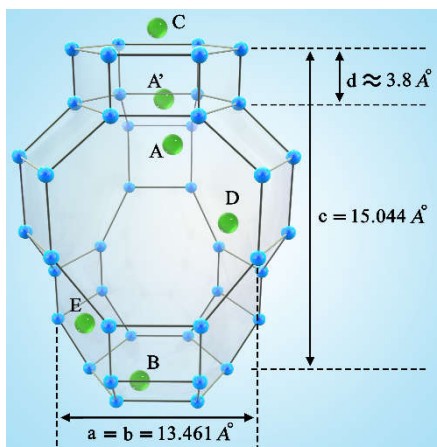

**Figure 2.** Schematic drawing of the SSZ-13 hexagonal unit cell structure and possible $Cu^{2+}$ locations (A, A′, B, C, D, E represent the possible $Cu^{2+}$ locations).

*2.1. NH₃-SCR Reaction Mechanism over the Cu-CHA Reaction*

Many researchers have been devoted to studying the SCR reaction cycle over Cu/SSZ-13, including the oxidation and reduction half-cycle, as shown in Figure 3a [21]. It has been reported that $NH_3$ can react with $[Cu^{II}(OH)]^+$ to form a $[(NH_3)_2-Cu^{II}-OH]^+$ complex that migrates into the CHA cage. NO adsorbed on $Cu^{II}$ and was oxidized to HONO, resulting in $Cu^{II}$ reduction to $Cu^I$. In the oxidation half-cycle at low temperatures (below 250 °C), $Cu^I$ and $Cu^{II}$ participated in the catalytic reaction in (the formation of pairing with $Cu^I(NH_3)_2$). It migrated to another $Cu^I(NH_3)_2$ and was activated by $O_2$ to produce a $Cu^I(NH_3)_2-O_2-[Cu^I(NH_3)_2$ dimer, and it was calculated by density functional theory (DFT) that $O_2$ activation was the rate-determining step of the $NH_3$-SCR reaction on Cu-SSZ-13 [21,22]. Palucci et al. calculated the diffusion radius of $Cu(NH_3)_2^+$ and found that it could migrate through the 8MR CHA [21]. More recently, Negri et al. discovered the formation of a $[Cu_2(NH_3)_4O_2]^{2+}$ (Figure 3b) complex with a side-on formation of a side-group (μ-η2, η2-peroxodiiamino dicopper) structure by X-ray absorption spectroscopy (EXAFS) and other measurements [22]. Meanwhile, their findings revealed that $[Cu_2(NH_3)_4O_2]^{2+}$ could be completely reduced to $[Cu^I(NH_3)_2]^+$ when NO and $NH_3$ appeared in the mixture [22]. These findings indicated a possible low-temperature SCR reaction mechanism of these complexes with NO. When the reaction temperature increased, isolated Cu ions species were anchored at the ion exchange site due to the decomposition of $Cu(NH_3)_n$ species [23–26]. Gao and his coworkers calculated that the activation energies of standard SCR at high and low temperatures were about 140 kJ/mol and 70 kJ/mol, respectively [5,23]. Janessens et al. [24] proposed the reaction mechanism of standard SCR and fast SCR, which was more like the reaction cycle at high temperatures, as shown in Figure 3c. According to this mechanism, the mixture of NO and $O_2$ was oxidized on the catalyst surface and subsequently reduced in the atmosphere of NO and $NH_3$. Contact of $Cu^+$ with $NO_2$ or with

a mixture of NO and $O_2$ could form the same dicoordinated (Cu-$NO_3^-$). In the presence of $NH_3$ and NO, $Cu^{2+}$ was reduced to $Cu^+$ and formed a product of $[Cu^I (NH_3)_2]^+$, which was a linear coordination of $Cu^+$ with $NH_3$.

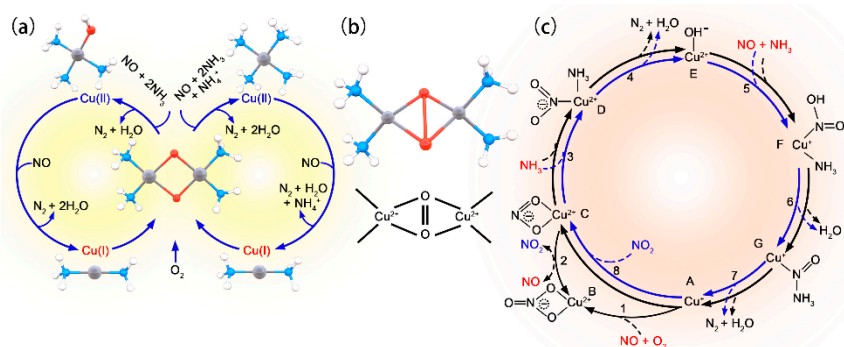

**Figure 3.** (**a**) Possible low-temperature SCR reaction cycle. Adapted with permission from Ref. [21]. Copyright 2017 Science. (**b**) μ-η$^2$,η$^2$-peroxo diamino dicopper (II) (side-on). Adapted with permission from Ref. [22]. Copyright 2020 American Chemical Society. (**c**) Possible high-temperature SCR reaction cycle. Adapted with permission from Ref. [24]. Copyright 2015 ACS Catalysts.

## 2.2. HTA Resistance over Cu-CHA

Due to the humid and high-temperature atmosphere in the ATS, researchers have been devoted to studying the HTA deactivation mechanism over Cu-CHA. Kwak [27] carried out studies on the HTA resistance over Cu-SSZ-13 and found that the lifetime exposure to exhaust emitted from diesel engines had little impact on its $NH_3$-SCR activity. Moreover, it has been reported that two types of active sites exist in Cu-SSZ-13, namely, the isolated $Cu^{2+}$ that resides in the six-membered ring and the hydroxylated $[CuOH]^+$ that resides in the eight-membered ring site [23,25,27]. Except for the Cu active sites, Brønsted acid sites functioning as $NH_3$ storage has also been reported as the rate-determining factor in $NH_3$-SCR of Cu-SSZ-13. Gao et al. comprehensively studied the hydrothermal aging mechanism over Cu/SSZ-13 [25]. It was reported that dealumination occurred and $Al(OH)_3$ formed, as shown in Figure 4a [25]. Moisture attacked the Si–OH–Al, resulting in the break of the Al–O bond. A further attack could completely dislodge the framework Al to form $Al(OH)_3$ outside of the framework. The dissociation of Al led to the absence of ion exchange sites and the unstable existence of $Cu^{2+}$ ions on the exchange sites. Cu aggregated to form CuOx species under high temperatures, as shown in Figure 4b [26]. Song et al. [26] found that $[CuOH]^+$ had lower hydrothermal stability than $Cu^{2+}$ at high temperatures, which indicated that Cu species residing in the eight-membered ring were more likely to aggregate to form $Cu(OH)_2$. Furthermore, $Cu(OH)_2$ could travel through the channels of Cu/SSZ-13, resulting in further destruction of the pore structure on the catalyst [24,26,27].

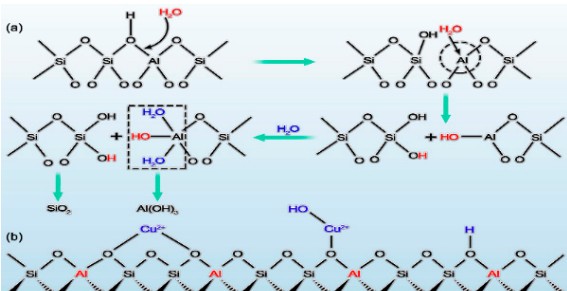

**Figure 4.** (**a**) Mechanism of dealumination over SSZ-13. Adapted with permission from Ref. [25]. Copyright 2015 ACS Catalysis. (**b**) Cu species reaction with water. Adapted with permission from Ref. [26]. Copyright 2017 ACS Catalysis.

## 3. Sulfur Poisoning and Regeneration over Cu/CHA

The sulfur poisoning over Cu-CHA is closely related to the $NH_3$-SCR mechanism. Meanwhile, the poisoning process is also closely related to the parameters of catalyst status, Cu loading, HTA, sulfurizing atmosphere ($SO_2$, $SO_3$, $SO_x$, $H_2O$, and $NH_3$), sulfur exposure time, exposure temperature, evaluation conditions, etc. [28,29].

### 3.1. SO₂ Poisoning over Cu/CHA

Great efforts have been made to study the impact of sulfur on catalysts. It has been proven by many groups that $SO_2$ severely inhibits the $NH_3$-SCR activity over Cu-CHA at low temperatures, while the high-temperature performance is nearly inhibited [25–30]. Wang et al. [30] verified that when Cu-SAPO-34 was exposed in $SO_x$ with different gaseous compositions, the NOx conversion at low temperatures was severely inhibited (100~300 °C) in a standard SCR reaction, as shown in Figure 5a. Furthermore, the TOF results showed that all the samples exhibited identical calculation values, which indicated that different $SO_x$ composition poisoning on Cu/SAPO-34 were all caused by the reduction of $Cu^{2+}$ quantity in the kinetic temperature range. In addition, for the remaining $Cu^{2+}$ that was not poisoned by $SO_x$, the activity remained unchanged [30].

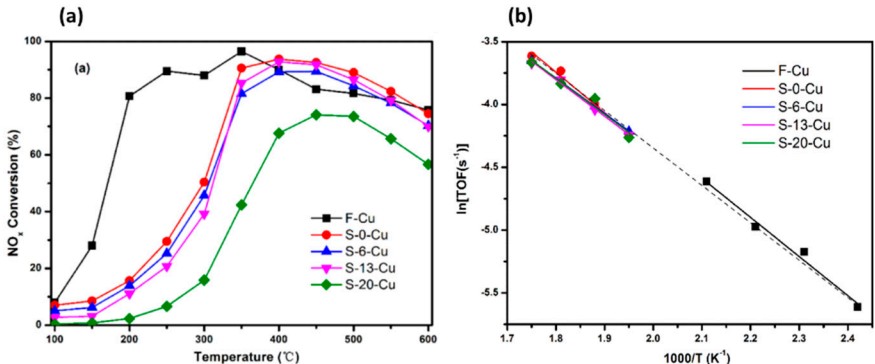

**Figure 5.** (**a**) NOx conversion of fresh and sulfated samples over Cu-SSZ-13; gas composition: 500 ppm NO, 500 ppm $NH_3$, 5% $O_2$, 7% $CO_2$, $N_2$. Adapted with permission from Ref. [30]. Copyright 2017 Applied Catalysis B: Environmental. (**b**) TOFs results over fresh and sulfated catalysts. Adapted with permission from Ref. [30]. Copyright 2017 Applied Catalysis B: Environmental.

Su [31] further compared the sulfate species on Cu-SAPO-34 and Cu-SSZ-13 and found that three main sulfates formed on zeolite, including $H_2SO_4$, $CuSO_4$, and $Al_2(SO_4)_3$. With stronger oxidation, more sulfates could be loaded on Cu-SSZ-13 compared with Cu-SAPO-34. By contrast, Cu-SSZ-13 and Cu-SAPO-34 tended to generate $H_2SO_4$ and $Al_2(SO_4)_3$, respectively. $H_2SO_4$ could be decomposed at a lower temperature than $Al_2(SO_4)_3$, resulting in the complete regeneration of Cu-SSZ-13 at 450 °C [30,31].

When evaluating the $NH_3$-SCR performance of Cu-SSZ-13 on a synthetic gas bench lower than 350 °C, the catalytic activity of sulfated samples was significantly decreased, indicating that most of the active sites were impacted by sulfur. When the reaction temperature reached beyond 350 °C, the impact of sulfur poisoning gradually decreased, which indicated the removal of sulfur substances and the recovery of active sites [31,32]. Luo et al. [32] revealed that both the $Cu^{2+}$ active sites coordinating with the six- and eight-membered rings were significantly reduced after sulfur exposure using an analysis of diffuse reflectance infrared Fourier transform spectroscopy (DRIFTs). The isolated $Cu^{2+}$ was supposed to react with sulfur, which resulted in a decrease in active sites and NOx conversion. Jangjou [33] reported that the decreased activity of Cu-SSZ-13 after sulfur exposure was related to the Cu-S species generated on the active sites. In addition, compared with isolated $Cu^{2+}$, active Cu $(OH)^+$ on the Cu/SSZ-13 framework was more likely to react with $SO_2$ to form sulfate. The sulfur exposure time, sulfur concentration, and exposure temperature of CHA were investigated by Ren et al. [34]. It was found that the sulfur loadings in the catalyst increased with an extension of the sulfur exposure time,

an increase in $SO_2$ content, and a decrease in exposure temperature. The relationship between sulfur content with a specific surface and the activity was investigated. Combined with thermo-gravimetric analysis (TGA) results, it was pointed out that pore blocks by ammonium sulfate and a decrease in the specific surface area on the framework directly led to deteriorated $NH_3$-SCR activity. Previous studies [33–35] observed $SO_2$ had a higher selective reactivity with $Cu^{II}$ species when it coordinated with $NH_3$ and extra-framework oxygen, in particular for $[Cu^{II}_2(NH_3)_4O_2]^{2+}$, by X-ray adsorption spectroscopy (XAS), as shown in Figure 6. By contrast, $SO_2$ was less reactive with Cu species in the absence of $NH_3$ or $[Cu^{II}_2(NH_3)_4O_2]^{2+}$, resulting in less $NH_3$-SCR deactivation under low temperatures [35].

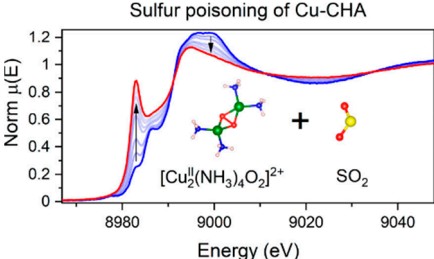

**Figure 6.** Cu K-edge XAS collected in situ during the exposure of the Cu species obtained to $SO_2$.

In a typical $NH_3$-SCR reaction, urea injection can be identified in the downstream sulfur atmosphere, and the co-adsorption of ammonia sulfate and Cu-S substance must be considered. Jangjou [33] found that when feeding gas containing a mixture of $SO_2$, $NH_3$, and NO, $NH_3$ facilitated $SO_2$ adsorption on Cu-SAPO-34 at 200 °C. Sulfur was stored on zeolite and co-adsorbed with $NH_3$, which resulted in the formation of ammonium sulfate. On the one hand, the DRIFT results showed that in the absence of $NH_3$, $Cu^{2+}$ resided at the six-membered ring and was completely poisoned by sulfur and partially poisoned at the eight-membered ring in a typical NO adsorption reaction [35,36]. On the other hand, the co-adsorbed ammonium sulfate could be consumed by NO in the SCR reaction [36]. Shen [37] proved that no matter which sulfate was generated at the active sites, the decrease in the active sites of isolated $Cu^{2+}$ mainly accounted for the decrease in activity. Wijayanti et al. [38] studied the sulfate species on Cu-CHA and found that $(NH_4)_2SO_4$ and $CuSO_4$ formed at 200 and 400 °C, respectively, on the isolated $Cu^{2+}$ active sites. Both led to deactivation over Cu-CHA [39,40]. It was proven that two kinds of active Cu species exist on Cu/CHA, namely $Cu^{2+}$-2Z and $[Cu (OH)^+]$-Z [40–43]. As shown in Figure 7, Jangjou [33] pointed out that $Cu^{2+}$-2Z had a different low-temperature $SO_2$ poisoning mechanism compared with $[Cu (OH)^+]$-Z. $Cu^{2+}$-2Z active sites were less reactive with $SO_2$. However, when $SO_2$ and $NH_3$ co-existed in a typical SCR reaction, $NH_3$ and $SO_2$ could react at a low temperature and decompose below 550 °C. On the contrary, $SO_2$ reactively adsorbed with $[Cu (OH)^+]$-Z and formed $CuHSO_3$, which could be composed below 580 °C. Above that, $CuHSO_3$ was oxidized to $CuHSO_4$ with gaseous oxygen feeding. Meanwhile, $CuHSO_4$ required a relatively high decomposition temperature, i.e., 750 °C, according to DFT calculation and FTIR measurement. In addition, this mechanism was further proven by Hannershoi [44] through quantitative identification of EPR. Olsson et al. proposed a kinetic model of $SO_2$ poisoning and regeneration in which the copper resided in the six-membered rings (S1Cu), and copper in the larger CHA cages (S2 and S3) adsorbed $SO_2$ and formed $S1_{Cu}$-$SO_2$ and $S2$-$SO_2$. In addition, $NH_3$ was adsorbed on those sites and formed $S1_{Cu}$-$SO_2$-$(NH_3)_2$ and $S2$-$SO_2$-$(NH_3)_2$ complexes, which worked as precursors of $(NH_4)_2SO_4$ [45].

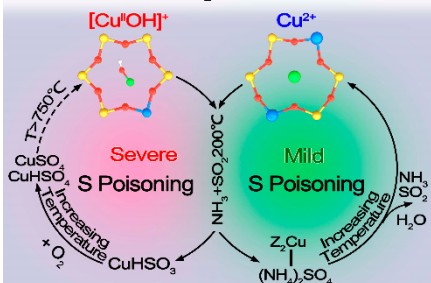

**Figure 7.** Low-temperature $SO_2$ poisoning mechanism. Adapted with permission from Ref. [33]. Copyright 2018 ACS Catalysis.

### 3.2. $SO_3$ Poisoning over Cu/CHA

In a typical ATS, a substantial percentage of $SO_2$ in the diesel exhausts can be oxidized by DOC. The impact of an upstream DOC catalyst on Cu-SSZ-13 was investigated by Kumar [46]. DOC was used for oxidizing CO/HC/soot and generating heat for the downstream DPF for the passive/active soot oxidization and fast reaction on SCR. During DOC oxidation, $SO_2$ tended to be oxidized into SOx, especially $SO_3$. Different sulfur species were added to the synthetic gas bench to study their influence on the $NH_3$-SCR activity of Cu-SAPO-34. The results showed that, compared with $SO_2$, $SO_3$ severely damaged the $NH_3$-SCR activity. Kumar also found that more copper sulfate could be generated under $SO_3$. Sulfated samples could not recover to the initial status after regeneration, inferring the formation of sulfate on Cu-SAPO-34. Wang et al. [30] pointed out that Cu-SSZ-13 can also oxidize $SO_2$ into $SO_3$ due to its strong oxidation, which can lead to the formation of $H_2SO_4$, $CuSO_4$, and $Al_2(SO_4)_3$ in a typical $NH_3$-SCR reaction. Kumar et al. [46] quantified the gaseous sulfur species on Cu-CHA zeolite and found that the $NH_3$-SCR performance was relevant to the sulfur loading and quantification. The FTIR measurement reported $SO_3$ transformation into $H_2SO_4$ in the $H_2O$ gaseous feeding. Cheng [47] evaluated the influence of $SO_2$ and $SO_3$ on Cu-zeolite and found that $SO_3$ had a more severe deterioration effect on $NH_3$-SCR performance. With identical sulfur content adsorption, it was easier for $SO_3$ to generate more sulfate. X-ray near edge spectroscopy (XANES) and X-ray photoelectron spectroscopy (XPS) techniques revealed that more $CuSO_4$ was generated under $SO_3$ exposure at the active sites compared with $SO_2$. $SO_3$ could result in a decrease in activity and could be recovered by the transformation of $CuSO_4$ into isolated $Cu^{2+}$ [48,49]. $H_2O$ also impacted sulfate formation on Cu-SSZ-13. Massive $CuSO_4$ was formed on Cu-SSZ-13 compared with moisture-free conditions in the feeding gas. Therefore, the SOx could react with single Al-coordination Z-Cu-OH without $H_2O$. By contrast, SOx could also react with dual Al coordination Z2-Cu with moisture that produced more copper sulfate, as shown in Equations (1) and (2) [41,49,50]. Shan's study also proved that $SO_2$ could dislodge the extra-framework Al that resulted from the dealumination during the hydrothermal aging process. More $Cu^{2+}$ active sites formed as CuOx after high-temperature sulfurization when compared with samples with only hydrothermal aging [50].

$$Z–Cu–OH + SO_X \rightarrow Z–Cu–HSO_{X+1} \tag{1}$$

$$Z2–Cu + SO_X + H_2O \rightarrow Z–OH–HSO_{X+1} + Z–H \tag{2}$$

It is worth noting that $NO_2$/NOx also has a significant impact on $NH_3$-SCR activity after exposure to sulfur. Compared with FSCR and slow SCR reactions, sulfated Cu/CHA experienced a drastic drop in NOx conversion in the standard SCR reaction but no obvious decrease in fast SCR and slow SCR reactions [28,29,51,52]. Wijayanti et al. [29] found that $(NH_4)_2SO_4$ and $CuSO_4$ formed under 200 and 400 °C, respectively, in a typical fast SCR reaction. In addition, sulfur deactivation was also closely related to Cu loading, Cu/Al, the aging degree, $H_2O$ gaseous feeding, etc.

On the basis of these studies, we found that the sulfate species and loadings generated under different sulfur exposure conditions were different and that the deactivation of $NH_3$-SCR activity over Cu/CHA was closely related to the decrease in active sites. Furthermore, hydrothermal aging typically took place when the catalyst was exposed in the humid atmosphere during DPF regeneration. It was pointed out that $SO_3$ can accelerate the dealumination on Cu-SSZ-13 and result in the irreversible collapse of the framework at high temperatures [50–54].

### 3.3. Sulfur Regeneration over Cu-CHA

The reversible and irreversible deactivation over Cu-CHA after sulfur poisoning has been investigated in recent years. In real applications, Cu-CHA regenerated along with the active regeneration over the DPF catalyst. Sulfates formed on Cu-CHA catalysts can be partially or completely restored during DPF active regeneration.

The regeneration mechanism over Cu-CHA varied with the feeding gas. Zhang et al. [40] reported that when both $SO_2$ and $NH_3$ were co-added into the $NH_3$-TPD, Cu-CHA was found with more ammonia stored. Ammonia sulfate was desorbed below 450 °C, indicating a relatively lower regeneration temperature. Su et al. [31] studied the sulfate regeneration of Cu/SSZ-13 catalyst and found that ammonium sulfide species could be removed at 450 °C, while copper sulfate needed a higher temperature before being completely removed.

Hammershøi et al. [44] studied the regeneration activity for the $SO_2$ poisoning of Cu-SAPO-34 and concluded that its $NH_3$-SCR activity could recover nearly 80% below 550 °C. When heated to 700 °C, it gained complete recovery. Wang et al. [30] reported the recovery mechanism of Cu-SSZ-13 in a wide temperature range. The phenomenon of reversible and irreversible deactivation was similar to that of Cu-SAPO-34. Wang explained the correlation between the regeneration temperature and SCR activity. It was reported that when the generation temperature was below 600 °C, there was a clear trend of irreversible deactivation over Cu-CHA, in a particular activity in a low temperature range [46]. Since $CuSO_4$ was decomposed to abundant CuO, which inhibited the SCR performance, the $NH_3$ oxidation increased at high temperatures [55–57]. When the regeneration temperature was increased to 650 °C, the deactivation at low temperatures declined significantly, which accounted for the $CuSO_4$ decomposition and migration to isolated $Cu^{2+}$ and $Cu^+$ [41–43]. These results were in agreement with studies of Shen et al. [55] and Jangjou et al. [33] regarding the recovery mechanism of stable copper sulfate decomposition on the Cu/SSZ-13 catalyst.

All in all, the relatively low $SO_2$ resistance limits the broader application of Cu-CHA in $NH_3$-SCR. Except for the regeneration method reviewed above, it was reported that some reductants, such as $NH_3$, $C_3H_6$, and n-$C_{12}H_{26}$, can achieve the removal of sulfate without heat treatment by changing the oxidation state of Cu [58–60]. Kumar et al. [46] reported that $1000 \times 10^{-6}$ $C_3H_6$ can achieve the complete removal of sulfate in the oxidizing atmosphere. It was proposed that recovery is beneficial for the reduction in binding energy between sulfate and Cu. Yu and his coworkers found that the hybrid catalysts incorporated ZnTiOx, which was proposed to work as a sacrificial reaction with $SO_2$, thus preventing the $Cu^{2+}$ from poisoning by $SO_2$ [61]. Other alkaline elements, such as Ce and Fe modification, can also improve the $SO_2$ resistance over Cu-CHA [59–61]. These findings enhance the $SO_2$ tolerance of Cu-CHA to some extent and give guidance to the further design of Cu-CHA in $NH_3$-SCR applications. However, the effect is still limited, and more studies are required in the future.

### 4. Conclusions

The deactivation and regeneration behavior over Cu-CHA in the $NH_3$-SCR reactions were reviewed in this paper. According to the findings reviewed above, it is obvious that different sulfate species and loading generated under different sulfur exposure condi-

tions can be identified. Sulfur poisoning and regeneration can proceed through different mechanisms depending on the conditions.

The sulfur poisoning that occurs at low temperatures over Cu-CHA by feeding $SO_2$ has an impact on SCR activity. Notably, $NH_3$ and $SO_2$ tend to form ammonium sulfate and coordinate with the isolated $Cu^{2+}$, which results in pore blocks and the deactivation of activity. Both copper sulfate and ammonia sulfate can impact $NH_3$-SCR activity by reducing the content of $Cu^{2+}$. In addition, the thermal decomposition temperatures of ammonium sulfate species and copper sulfate are around 550 °C and 700 °C, respectively. Meanwhile, the regeneration process is always accompanied by the redispersion of copper. Therefore, the activity caused by $SO_2$ or ammonium sulfate species could be partially or completely recovered.

$SO_3$ is reported to have more severe deactivation within 200~400 °C compared with $SO_2$, which is mainly attributed to the massive $CuSO_4$ formation. The presence of $SO_3$ could cause dealumination on the framework, resulting in the fracture of the Si–O(H)–Al bond. In addition, $SO_3$ can poison more isolated $Cu^{2+}$; therefore, the decrease in $Cu^{2+}$ is the main reason for the decrease in $NH_3$-SCR activity. Furthermore, the decrease in $Cu^{2+}$ and the fracture of the Si–O(H)–Al bond lead to a drop in high-temperature activity. The formation of copper sulfate requires nearly 700 °C for regeneration. It is always accompanied by the formation of CuO and Cu migration in the cage. However, the active sites cannot completely recover because of the influence of dealumination.

It can be seen that many trials have been carried out to improve the $SO_2$ tolerance over Cu-CHA. The addition of other metals as a sacrificial component is proposed as an efficient way to improve its $SO_2$ tolerance in real applications. Moreover, it is essential to clarify the nature of co-poisoning by alkaline, $PO_4^{3-}$, etc. with sulfur in real applications. This could provide a specified regeneration strategy for calibration over the ATS, thus enhancing the durability and NOx abatement of Cu-CHA in industrial applications.

**Author Contributions:** Conceptualization, J.M. and H.L. investigation, J.M. and F.Y.; writing—review and editing, J.M. and S.C. Chang; supervision, Y.Z.; project administration, Y.Z. All authors have read and agreed to the published version of the manuscript.

**Funding:** This research was funded by the Provincial Key S&T Special Projects of Yunnan (grant number 202102AB080007) and the National Key Research and Development Program of China (grant number 2021YFB35033200).

**Conflicts of Interest:** The authors declare no conflict of interest. The funders had no role in the design of the study; in the collection, analyses, or interpretation of data; in the writing of the manuscript; or in the decision to publish the results.

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
