# Peer review of "Research Progress on Sulfur Deactivation and Regeneration over Cu-CHA Zeolite Catalyst"

_catalysts, doi:10.3390/catal12121499_

Round 1

Reviewer 1 Report

The submitted manuscript reviews the effects of sulfur poisoning on the deactivation of Cu-Chabazite catalysts and the reported actions for regeneration of the catalysts towards the NOx reduction applications in NH3-SCR process. I think this is a worthy contribution to the literature however, there are several issues and missing literature that need to be addressed and incorporated before publication.

Although most of the commercial applications are using Cu-SSZ-13, a significant section of the review is on SAPO-34 samples which actually lacks durability at low temperature upon contact with moisture (Nature Communications volume 10, Article number: 1137 (2019)). This needs to be addressed and the manuscript should be re-organized to reflect this.

Effects of hydrothermal aging on Cu-SSZ-13 should also be discussed in tandem with the effects of the sulfur poisoning (i.e. effects on ZCuOH and Z2Cu sites, and also on the SCR performance).

The sentence “In addition, NO2 and N2O concentrations produced are all lower than 10ppm during the whole SCR reactions” in Section 2 is not clear to me. This does not seem to be correct if different ANR or feed NOx ratios are considered.

The reasons behind the good HTA durability of Cu-SSZ-13 should also be discussed along with sulfur poisoning effects.

Kwak instead of Kwark in Section 2.

The paragraph on the formation of active species during catalyst preparation should incorporate different scientific views too. Works by Paolucci et al., Yezerets et al. and Tronconi et al. need to be incorporated.

Figure 4. This is an XAS data, please give the correct caption. It is not clear for the common reader’s eye how the changes in XAS data (i.e. changes in the oxidation state) is associated with SO2 influence on SCR.

Generally, in this manuscript, the influence of SOx poisoning is discussed in a qualitative manner. It would be good to see how much Standard SCR or Fast SCR conversion, reaction rates or TOFs are decreasing upon sulfur poisoning wrt temperature at a certain space velocity.

Authors should first describe what DPF active regen is before discussing its effects on SCR sulfur regeneration.

English of the manuscript requires significant improvement. For example, the expression “law of ammonium sulfate decomposition” should be changed. There is no such a thing as a law of ammonium sulfate decomposition.

There are a few good articles on the modeling of Sulfur poisoning and regeneration which need to be incorporated to the article such as Olsson et al., Epling et al. and Cummins group articles.

It is hard to understand what is happening on Fig. 6. More clarification and discussion is needed in the text.

As far as I understand, most severe S-poisoning occurs due to SO3 poisoning instead of SO2 poisoning. This needs to be addressed and discussed with quantitative data from the literature.

Author Response

Please find the point-by-point responses attached.

Reviewer 2 Report

This is a good review summarizing the research progress on sulfur deactivation and regeneration over Cu-CHA zeolite catalyst. It is a significant issue to resolve the deactivation of Cu zeolite catalyst for SCR reaction. However, parts could be updated and polished to make this manuscript published. 

Comment I: In the introduction, it would be more impacted if authors make a scheme/graph generally demonstrating the SCR reaction through Cu-CHA catalyst. And list the common issues cause the deactivation of Cu sites or Cu-CHA type of catalysts. Authors summarize in the paragraph of introduction but graphic abstract is clearer. 

Comment II: In section 3, authors could list subsections of each reason/mechanism of sulfur poisoning, and correspondingly illustrate the approach to regenerate or recover the active sites. The logic of this section would be more smooth.

Comment III: In conclusion, authors could also provide a perspective of such Cu-zeolite avoiding deactivation/or better regeneration for the future research.

Comment IV: resolution of figures in manuscript seems not very qualified. Authors should update to high resolution pictures and cite the original reference if you reprint from other papers. 

Author Response

(The authors gave the same response as above.)

Reviewer 3 Report

The reviewed article reviews the deactivation and regeneration behavior of Cu-CHA in NH3-SCR reactions.  Sulfur poisoning and regeneration can follow different mechanisms depending on the conditions. In addition, sulfur poisoning in Cu-CHA is closely related to its composition, exposure temperature and exposure time. NH3-SCR activity can fully or partially recover with  sulfur removal and sulfate decomposition. Ammonia sulfate can be completely decomposed below 550℃, while Cu-S species required a regeneration temperature higher than  than 600℃. Cu-S species can be partially or completely regenerated to Cu2+ and Cu+ , accompanied by removal of sulfur substances and recovery of active sites.

The reviewed article is very interesting. Each of the presented parts of the article was developed and presented in a very clear way. The entire article is coherent, closely related and presents results of high scientific value. In summary, each part of the publication has been described in detail, the charts, graphs presented are very clear to the audience. The conclusions are consistent and closely related to the research topic. As a reviewer of this work, I will not make any comments. I believe that the authors have exhausted all the topics contained in the reviewed work.

Author Response

(The authors gave the same response as above.)

Reviewer 4 Report

The submitted manuscript focuses on clarifying the deactivation and recovery mechanism over Cu-CHA, including active sites, NH3-SCR reaction mechanism, law of sulfur poisoning, and recovery mechanism over its structures. The manuscript is well prepared and meets the requirements outlined in the Journal of Catalysts. However, the lines numbers are missing from the manuscript, thus making difficult the review process. Overall, I think that the presented work is attractive and the manuscript has enough scientific content. As such, I would suggest its publication after minor revisions as shown below:

- Abstract:  The abstract summarizes the research, but needs to be reworked to be clearer and more precise.  Please also explain to readers what new things were discovered in this review study. What is the big conclusion?   It is very important to state how this study will improve the existing literature in this section.

- The novelty of the study should be more highlighted. What is novel/new in this study, compared with other similar studies? These things must be clearly explained in Introduction.

- The conclusions section is extremely weak - basically just re-opens the arguments, and doesn't effectively bring out the key points from the main body of the text. Thus, the review does not really provide a way forward and remains only a repetition of the abstract.

Other comments
-Please expand abbr. when first seen in the text. E.g. SCR in the abstract

- In most cases figures are of poor quality.

-There seems to be simple message that is lost in the jumble- p.7. “This also gives insights into subsequent deSOx studies and real applications”. On what basis? Please be more specific herein.

- p.4. Please change Ren into “Ren et al.,”

Author Response

(The authors gave the same response as above.)
